# The Flexible Operation of Coal Power and Its Renewable Integration Potential in China

**Chunning Na [1],\*** , **Huan Pan [1]** , **Yuhong Zhu [1]** , **Jiahai Yuan [2]** , **Lixia Ding [3]** **and Jungang Yu [4]**

[1]   School of Physics and Electrical Engineering, Ningxia University, Yinchuan 750021, China
[2]   School of Economics and Management, North China Electric Power University, Beijing 102206, China
[3]   Eco-tech Research Institute, State Grid Ningxia Electric Power Co., Ltd., Yinchuan 750004, China
[4]   Ningxia Jingneng Ningdong Electric Power Co., Ltd., Yinchuan 750409, China
\*   Correspondence: nana508@163.com

**Abstract:** At present time, China's power systems face significant challenges in integrating large-scale renewable energy and reducing the curtailed renewable energy. In order to avoid the curtailment of renewable energy, the power systems need significant flexibility requirements in China. In regions where coal is still heavily relied upon for generating electricity, the flexible operations of coal power units will be the most feasible option to face these challenges. The study first focused on the reasons why the flexible operation of existing coal power units would potentially promote the integration of renewable energy in China and then reviewed the impacts on the performance levels of the units. A simple flexibility operation model was constructed to estimate the integration potential with the existing coal power units under several different scenarios. This study's simulation results revealed that the existing retrofitted coal power units could provide flexibility in the promotion of the integration of renewable energy in a certain extent. However, the integration potential increment of 20% of the rated power for the coal power units was found to be lower than that of 30% of the rated power. Therefore, by considering the performance impacts of the coal power units with low performances in load operations, it was considered to not be economical for those units to operate at lower than 30% of the rated power. It was believed that once the capacity share of the renewable energy had achieved a continuously growing trend, the existing coal power units would fail to meet the flexibility requirements. Therefore, it was recommended in this study that other flexible resources should be deployed in the power systems for the purpose of reducing the curtailment of renewable energy. Furthermore, based on this study's obtained evidence, in order to realize a power system with high proportions of renewable energy, China should strive to establish a power system with adequate flexible resources in the future.

**Keywords:** coal power; flexible operation; renewable energy; integration potential; China

## 1. Introduction

The integration of renewable energy in China's power system is in urgent demand in order to promote a sustainable energy transition [1]. During the last decade, technologies related to renewable energy have experienced exploding increases. By the end of 2018, China enjoyed the largest scale wind power energy resources in the world, with an installed capacity of 184 GW [2]. However, China still faces enormous challenges in regard to integrating large-scale renewable energy into the current power system while reducing curtailed renewable energy [3]. In order to avoid curtailment and promote the integration of renewable energy processes, the flexibility requirement of China's power systems will be required to reach higher levels [4]. Therefore, various flexible resources have been proposed in order to achieve these flexibility requirement goals. These include interregional transmissions [4],

energy storage technology [5–7], and demand responses [8–10]. Meanwhile the availability of the aforementioned resources is currently not sufficient to meet the flexibility requirements of China's power system [3]. Due to China's existing coal power endowment, the flexible operation of the existing coal power units is still the most feasible choice to ensure the integration of renewable energy, and would potentially contribute to reductions in the curtailment of renewable energy, as well as ensure the safe, stable, and reliable operation of the power grids [11].

Currently, a tremendous amount of research is being focused on the evaluation indexes and models of power system flexibility [12–19]. However, not as many research studies are presently exploring the flexibility potential of coal power based on simulations or modeling methods. Kubik et al. highlighted that insufficient attention has been given to the potential flexibility of the existing thermal plants, and put forward certain operating strategies with three thermal units for the purpose of reducing variability impacts using a unit-by-unit approach [20]. However, it has been suggested that this would be an inefficient approach for larger energy systems, and closer attention should be paid to the units' characteristics and non-synchronous generation limits. In the studies conducted by Stefanía et al., it was also indicated that coal-fired power generators could potentially supply flexibility and contribute to lower system costs in three regional electricity generation systems using a linear cost-minimizing investment model [21]. However, it was found that both of the aforementioned studies had not calculated the potential flexibility of thermal power. In another related study, Luo et al. has predicted the minimum technical output of thermal power under different levels of accommodating wind power in typical days of 2020 in North Hebei (China), and stated that the minimum technical output of condensing units must be lower than the international advanced level of 25%, in order to achieve the objective of a 10% curtailment rate of renewable energy [11]. However, the performance impacts of coal power units during flexible operations [22] and the operation economy were not considered in the aforementioned study. Also, in the operational model, it was found that both the joint operations with energy storage technology and the flexible operations with different ramp rates had not been taken into consideration.

In the present study, the performance impacts and the operation economy were first reviewed. Then the flexible operation models of the peak shaving and ramp rates were established. Following this, the promotion effects of the renewable energy's integration based on a Mixed-Integer Programming model was proposed, in which the production processes with the existing coal power units flexible operations were simulated during the course of a one-day period. The main contributions of this study were from three aspects. The first was the systematic explanation of the reasons for continuing to depend on the existing coal power units in order to promote the integration of intermittent renewable energy in China. The second aspect involved a review of the negative impacts of flexible operation on coal power units. The third was the construction of a flexible operation model, and the calculation of the integration potential using sequential production simulations. Then, the boundary conditions of the renewable capacity could be deduced using the constant coal power capacity and different peak shaving depths of the coal power units.

The research study is organized as follows: Section 2 reviews the reasons for the use of the flexible operations of coal power units to promote the integration of renewable energy in China. Section 3 describes the impacts on the flexible operations of the coal power units on their future performances. Section 4 introduces this study's constructed simulation model and various case scenarios. Also, this study's calculations of the potential of the promoted renewable energy provided by the flexible operations of the coal power in different scenarios are detailed. Finally, Section 5 summarizes the conclusions reached in this study.

## 2. Reasons for the Flexible Operation of China's Existing Coal Power Units in the Promotion of the Integration of Renewable Energy

### 2.1. Coal Power Remains the Main Generation Power

Although the installed capacity share of coal power has dropped year by year, it still remains the main generation power source in China. As shown in Figure 1, by the end of 2018, the share of coal power (53%) had surpassed more than half of the total installed capacity, while its current feed-in tariff was lower than that of the other types of power generation [23]. Meanwhile, it can be seen in the figure that its development trend had changed from ensuring a power supply to both ensuing a power supply and providing the ancillary services for renewable energy [1]. However, due to the plague of overcapacity in coal power [24,25], its capacity in China will not exceed 110 GW by 2020 [26]. Therefore, under the double pressures of environmental pollution and climate change, the power systems must rely on the flexible operation of the existing coal power units in order rapidly respond to the residual load changes, with the purpose of promoting renewable energy integration.

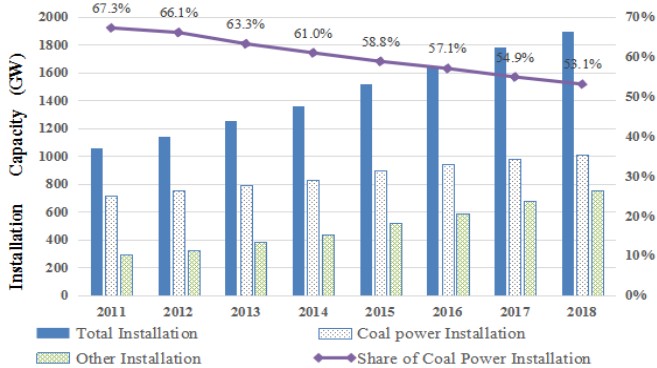

**Figure 1.** Installed capacity and share of coal power from 2011 to 2018 [27–29].

Figure 2. Displays the unit sizes of the thermal power units and the corresponding capacities as of 2016.

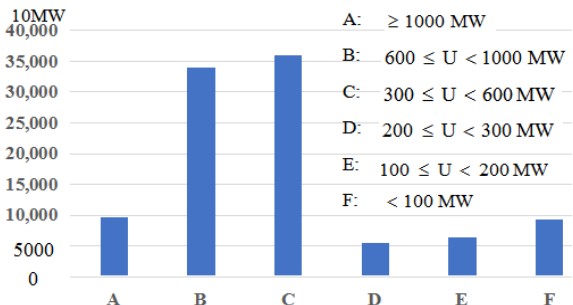

**Figure 2.** China's thermal power units in 2016 [30].

In China, 90% of the thermal power units are coal power units. Additionally, the units in the 300 MW to 600 MW range are the mainstream units, with a capacity share of 37%. Meanwhile, they are generally considered to be the sub-critical units within the category of the flexible operation units. The units in the 600 MW to 1000 MW range are second, with a capacity share of 32%. As of the end of 2016, there were 100 units in 1000 MW, with a capacity share of 9.5%. The other small-scale units accounted for only a minimal capacity share. Among those, the units under 200 MW will be phased out in the next two to three years [26], following consideration being given to their low efficiency and high pollution effects.

## 2.2. High Costs and Severe Capacity Shortages in Natural Gas Power

Although the minimum steady output of a gas power unit is in the range of 0% to 30%, the ramp rate ranges between 7% and 30% of the rated power every minute, and the start-up time is usually within 10 min to one hour [31]. All these operational parameters are much better than those of the generally flexible coal power units [31], which means that gas power units are essentially more flexible than coal power units. However, in regard to the resource endowments, the available reserve resources of coal remains rich, while that of natural gas is currently poor, which has contributed to the higher prices of natural gas in China. Take Zhejiang Province as an example, the fit in the tariff of natural gas power is 96.345 $/MWh [32], which is more expensive than that of coal power (60.672 $/MWh) [23].

In addition, the capacity share of the gas power units in China was only 4.4% at the end of 2018 (Figure 3), whereas the total capacity shares of wind power and solar power had significantly increased to 19% during that year [29]. Additionally, from the viewpoint of installation size, it is clear that the existing gas power units in China are insufficient to supply the needed operational flexibility and also bear the heavy burden of promoting the integration of renewable energy. Miguel et al. also revealed that at their minimum load rates, gas power plants are less flexible and produced more NOx and CO emissions than coal power plants [33]. Therefore, when considering the operation economy, gas power is usually considered to be a base load and unfit for low load operations with the purposes of promoting renewable energy integration.

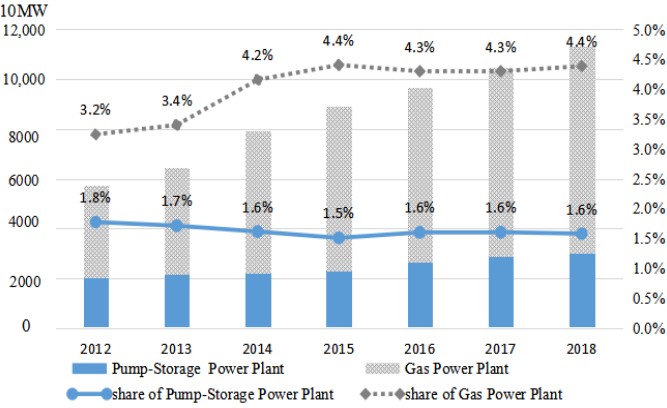

**Figure 3.** Installed capacities and shares of Pump-Storage Power (PSP) and gas power plants [27–29].

## 2.3. Capacity Shortages in Pump-Storage Power (PSP) Plants

PSP plants is a type of special power supplier with two main features. On the one hand, they are both a power supplier and a power user, with the ability to store energy in the load valleys, while outputting power in peak loads. On the other hand, they are considered to be the most effective accident cold standby units, with an outstanding short operation times of between 120 s and 150 s from a standby state to a rated power state [31]. Moreover, the PSP plants can operate flexibly and reliably to quickly respond to the load changes [34,35], which is superior to any other thermal power units [34]. Additionally, the PSP plants' abilities in peak shaving are two times their capacities [36], which is much higher than all of the conventional units [31]. Although the PSP plants have the lowest levelized cost of energy [37] and the acceptance of renewable random fluctuations to a certain extent [38–41], the cumulative installed capacities have been less than 2% of the total generation capacity in recent years (Figure 3), which is far less than the capacity size of renewable energy [27–29] in China.

It has been found that, due to the influences of the geographical environments, it is not suitable for the PSP plants to be constructed in water resource constraint regions [22]. Therefore, the installed capacity had increased from 2.33 GW in 2012 to 2.99 GW in 2018, and the capacity share of the PSP plant had still remained lower than 2%. As a result, the existing PSP plants can no longer accommodate the heavy burden of promoting the integration of renewable energy under the conditions of the current

installed capacity and geographical restrictions. Therefore, it will be necessary for coal power units to undertake more long-term peak shaving tasks [22].

*2.4. Small Capacities and High Commercial Costs of Energy Storage Devices*

Along with PSP plants, there are several other energy storage (ES) technologies, like compressed air (CA), batteries (BA), flywheels (FW), superconducting magnetic (SM), and super-capacitors (SCP). Table 1 lists the performances comparison of several ES technologies. The rated power of the PSP plants ranges between 100 MW to 5000 MW, which is much higher than other ES technologies. The highest efficiency rate is 87%, which is not the best of all of the ES technologies. However, the PSP plants have the longest spans (60 years), while that of the SCP is only 5 years. Also, the investment costs are almost 10 times the investment costs of the SM, but the variable operational and maintenance (O&M) costs are 0.5 $/kW, only one-fortieth that of the SM. Therefore, among the aforementioned energy storage technologies, the PSP plants are the most mature technology, with the largest rated power, longest life span and the lowest O&M costs.

**Table 1.** Of the performances of several energy storage (ES) technologies.

| | PSP | CA | BA | FW | SCP | SM |
|---|---|---|---|---|---|---|
| Rated Power (MW) | 100 to 5000 | 100 to 300 | 0.001 to 50 | 0.005 to 1.5 | 0.001 to 0.01 | 0.010 to 1 |
| Efficiency Rate (%) | 65% to 87% | 40% to 75% | 70% to 90% | 90% to 93% | 60% to 90% | 80% to 90% |
| Life Span (year) | 40 to 60 | 30 to 40 | 3 to 15 | 0 to 15 | 0 to 5 | >20 |
| Investment Cost ($/kW) | 1500 to 3000 | 550 to 1250 | 1085 to 2500 | 350 | 300 | 300 |
| Fixed Operational and Maintenance (O&M) Cost ($/kW) | 5 | 15 | 250 to 800 | 7.5 | 5.5 | 25 |
| Variable O&M Cost ($/kW) | 0.5 | 1.7 | 1.0 | 4 | 5 | 20 |

Note: Data were sorted by the authors according to [7,31].

It has been found that although the BA, FW, SCP, and SM have high efficiency rates, their rated power and life span are quite low [7]. For example, the maximum rated power of the SCP is only 0.01 MW, which is only one-millionth that of the capacity of the smallest PSP plant, with 10 times the variable O&M costs of the PSP plants. Furthermore, with the exception of the PSP, these ES technologies can be used in an infinite loop with high efficiency [7], but have the disadvantages of small capacities, unsafe operations, and high commercial costs. This is particularly true for BA, which have the highest investment costs and fixed O&M costs. Therefore, in recent years, they have not been widely used commercially for smoothing power fluctuations or promoting the integration of renewable energy.

## 3. Impacts of the Flexible Operation on the Performances of the Existing Coal Power Units

Generally speaking, coal power units are used as the base load power plants in China. Due to the growing scale and the severe curtailment of renewable energy, the utilization of the existing coal power units as peaking power plants has become an inevitable trend, particularly in areas where no PSP plants have been deployed. The performance rates of the generally operated coal power units, such as start-up times, ramp rates, and the minimum output, have been determined to be unsuitable for peaking power plants, which has made the flexibility retrofitting of the existing coal power units an imperative task.

The flexible operations of coal power units usually refers to coal power units running in a frequent start–stop or low load operational modes after retrofitting, which tends to dramatically shorten their life spans. In addition, the low-load operations would also inevitably promote their own consumption rates and coal consumption rates [21,22]. These two parameters in different load factors with different units are shown in Table 2 and Figure 4, respectively. As can be seen in Table 2, the own consumption rates reflected the consumed electricity of all the electrical equipment in the plant within a specified period under the normal operational circumstances. When the coal power units run at a rated power, they enjoy the lowest own consumption rate. However, with the reduced output required to give the generation space to renewable energy, it can be seen that the own consumption rates soared. It was

observed that the greater the reduced output was, the higher the increment would be. In particular, as the ultra super-critical (USC) units ran at lower than 50% of rated power, their own consumption rate and coal consumption had clearly soared, with increases of approximately 1.2 percentages and 60 g/kWh in 30% rated power, respectively. As a result of these conditions, the high efficiency and energy saving have been lowered. Also, when compared with the USC units, the impacts of the low load operations on the performance of the super-critical (SC) units and sub-critical (SBC) units were found to be much smaller. Therefore, USC units are usually not considered as the flexibility retrofitting units, but rather as the base load power plants with full-power operations.

**Table 2.** Own consumption rates in different load ratios. USC: ultra super-critical; SC: super-critical; SBC: sub-critical.

| Capacity | | Features | Own Consumption Rates | | | | Increment | | |
|---|---|---|---|---|---|---|---|---|---|
| | | Load ratios | 100% | 50% | 40% | 30% | 100% to 50% | 100% to 40% | 100% to 30% |
| 1000 MW | A | USC + water cooled | 4.16 | 4.93 | 5.14 | 5.33 | 0.77 | 0.98 | 1.17 |
| 660 MW | B | USC + water cooled | 4.26 | 4.84 | 5.02 | 5.21 | 0.58 | 0.76 | 0.95 |
| | C | SC + water cooled | 4.65 | 5.19 | 5.39 | 5.57 | 0.54 | 0.74 | 0.92 |
| | D | SC + air cooled | 4.83 | 5.40 | 5.57 | 5.73 | 0.57 | 0.74 | 0.90 |
| 600 MW | E | SBC + water cooled | 4.79 | 5.30 | 5.47 | 5.66 | 0.51 | 0.68 | 0.87 |
| | F | SBC + air cooled | 5.20 | 5.82 | 6.01 | 6.22 | 0.62 | 0.81 | 1.02 |
| 300 MW | G | SBC + water cooled | 4.67 | 5.24 | 5.43 | 5.61 | 0.57 | 0.76 | 0.94 |
| | H | SBC + air cooled | 5.56 | 6.17 | 6.36 | 6.57 | 0.61 | 0.80 | 1.01 |

Source: The predicted economic index obtained from the Eco-tech Research Institute of State Grid Ningxia Electric Power Co., Ltd. according to the units of the entire power sector.

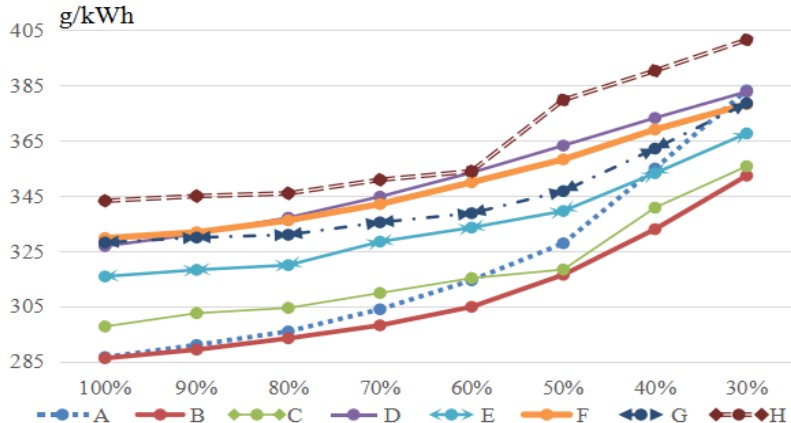

**Figure 4.** Power supply coal consumption rates with different units in different load factors. Source: The predicted economic index obtained from the Eco-tech Research Institute of State Grid Ningxia Electric Power Co., Ltd. according to the units of the entire power sector.

It has been found that low load operations also have impacts on flue gas denitration systems in coal power plants. Since the systems use the Selective Catalytic Reduction (SCR) technology, the reaction temperatures commonly range between 320 to 400 °C. However, the flue gas outlet temperatures will be below the reaction temperatures due to the low load operations of the units, which will lower the safety degree of the system, and increase the air pollution emissions. For instance, the 660 MW·SC air-cooled units of the Jinneng Ningxia Plant are currently at the low-load running state of 30% of the rated power with a flue gas outlet temperature of 295 °C. Meanwhile the inlet flue gas temperature will reach that of the reaction temperature after the flexibility retrofitting.

However, it has been found that the amounts of generated NOx are higher in low load operations. Therefore, in order to improve this situation, the plant operators have reduced the amount of oxygen in the furnace chambers to a certain level. Unfortunately, if excessive adjustments occur, the furnace

chambers may experience unstable combustion situations, which would in turn to lower the efficiency of the SCR reactor operations.

Moreover, the released $NH_3$ from the SCR reactors will react with $SO_3$ and $H_2O$ in the flue gases to produce $NH_4HSO_4$ at a certain temperature. This strong cohesiveness product will adhere to the surfaces of the air preheaters and lead to the accumulation of fly-ash particles. Then, after long periods of time, these accumulated ash particles will increase the air-cooling fans' power consumption, which would endanger the safe operations of the units, and even shorten the life spans of the units.

Such flexibility operations as high ramp rates, low load operations and frequent start–stop actions have major negative impacts on the life spans of units [22,42,43]. The life attrition is dependent on the changes of the temperature and pressure of the furnace, as well as other factors. A coal power plant was simulated with a 50-times annual start–stop rate and a 2-times normal ramp rate. It was found that its attrition rate increased from 0.4% to 3.24%, which was 8 times of the original value [43]. These results indicated that a unit with a designed 30-years life-span had a reduced lifespan of 0.97 years under the simulated conditions. The lifespan distributed data for a unit of the General Electric (GE) Co. are shown in Table 3 [44]. As can be seen in the table, the frequent start–stop mode had increased the furnace's attrition and reduced the unit's lifespan. The operation of the cool start-up had the highest attrition rate of 0.15%. Also, the attrition rate of the start-up after stopping 2 h was 5 times that of the rate after stopping for 2 days. Meanwhile, the start-up after stopping 8 h to 2 days had the same attrition rate. For example, for units with the same designed lifespan, the service life had decreased 2.25 years after 50-times cool start-ups. As the units operated flexibly with a rate of 50 times for the start-up after 2 h of stopping, its service life had decreased by 0.75 years. In particular, the attrition rates of the major great amplitude variable loads were 40 times that of the rate with small amplitude variable loads, which indicated that a unit operating with a 50-times severe peak load shaving would have its service life decreased by approximately 0.15 years. Then, if the unit operated 500 times annually under the aforementioned conditions, its lifespan would be 5 years rather than 20 years, which meant that the unit would be almost approaching retirement after 10-years of service.

**Table 3.** Lifetime distributed data for a unit of General Electric Co.

| Operation Mode | Attrition Rate (%) | Cumulative Running Times | Lifetime Attrition (%) |
|---|---|---|---|
| Cool Start-up | 0.15 | 100 | 15 |
| Start-up after Stopping 2 Days | 0.01 | 500 | 5 |
| Start-up after Stopping 8 h | 0.01 | 200 | 2 |
| Start-up after Stopping 2 h | 0.05 | 100 | 5 |
| Major Amplitude Variable Load | 0.01 | 300 | 3 |
| Small Amplitude Variable Load | 0.00025 | 10,000 | 2.5 |

Low load operations will incur increasing own consumption rates, as well as increased coal consumption of the units, and also have negative impacts on the flue gas denitration systems and the lifespans of the units. From an economic viewpoint of the entire power system, the units with flexible operations would tend to promote the integration of renewable energy and decrease their power outputs, thereby saving on the cost of fossil fuels and reducing the total emissions of $CO_2$ and other pollutants. Furthermore, due to the much higher contributions in saved costs when compared with increased O&M costs in coal power plants, the flexibility operations with markets designed for such operations would be profitable [42]. Therefore, improving the operation flexibility of coal power is undoubtedly an important and feasible option to further increase the proportion of renewable energy in China. Such steps will be beneficial for ameliorating the current situation of heavy coal-dependent electricity.

## 4. Promotion of the Integration of Renewable Energy through the Flexible Operation of Existing Coal Power Units

This section details this study's sequential production simulations of an IEEE 10 units/39 nodes system with wind power (WP) and photovoltaic power (PV). These simulations were under taken in

order to investigate the promotion of renewable energy with the flexible operation of coal power units by employing mixed integer programming (MIP) method.

MIP is a common method of modeling production and business activities related to specialized issues, such how to utilize resource effectively. Generally speaking, the models are used simulate the operations of power systems in competitive electricity markets [45]. During the simulation processes certain constraints are adopted and there are several commitments by which the units operate. The output of those operating units is determined with a certain object function. However, despite the known advantages of the MIP method, it requires the solving of an NP (non-deterministic polynomial)-hard problem [46]. Once a power system composed of several thousands of power plants is modeled, it would take an enormous computational time to model the detailed behaviors of the individual plants, including many constraint conditions. Therefore, in order to reduce the complexity problem, an IEEE 10 units/39 nodes system was adopted in different scenarios in this research study.

The purpose of the aforementioned simulations was to solve two issues. The first investigated issue was the scale of renewable energy which could potentially be consumed with the flexible operation of existing coal power units. The second issue was the determination of how much of the increment scale of the renewable energy could be potentially consumed with the peak shaving depths changing from 38% to 20% of the rated power. In the present study, the maximum consumption levels of WP and PV were considered as the object functions using Formula (1) as follows:

$$F = \max \sum_{t=1}^{T} (P_w^t + P_v^t) \tag{1}$$

where $P_w^t, P_v^t$ represent the integrated WP and PV in the moment of $t$, respectively. Then following the flexibility retrofitting, with the units running with higher peak shaving depths, the output constraints will be changed into flexible constraints as indicated in Formula (2).

$$C_j^t(P_j^{\min} - \Delta P) \leq P_j^t \leq C_j^t P_j^{\max} \tag{2}$$

where $j$ is the unit number; $P_j^t$ indicates the output at the moment of $t$ for unit $j$; $P_j^{\min}, P_j^{\max}$ are the lower and upper output boundaries, respectively; and $C_j^t$ represents a binary variable which indicates whether the unit $j$ is running at moment $t$.

As shown in Formula (3), the ramp rate constraints are usually used in dealing with dispatching units schedules for thermal power systems [47], and can be applied to improve the reliability of unit commitments. This is particularly important for the combined optimization of thermal power and PSP with intermittent energy, in order to improve operational security and promote the utilization of intermittent energy [39]:

$$\begin{cases} P_j^{t+1} - P_j^t \leq P_j^{up}\left(1 + C_j^t - C_j^{t+1}\right) + P_j^{\min}(2 - C_j^t - C_j^{t-1}) \\ P_j^t - P_j^{t+1} \leq P_j^{down}(1 - C_j^t + C_j^{t+1}) + P_j^{\min}(2 - C_j^t - C_j^{t-1}) \end{cases} \tag{3}$$

Then, following the flexibility retrofitting, the improved ramp rates will become higher than the designed ramp rates. Therefore, their constraints will change into flexible constraints, as shown in Formula (4):

$$\begin{cases} P_j^{t+1} - P_j^t \leq (P_j^{up} + \Delta P^{up})\left(1 + C_j^t - C_j^{t+1}\right) + P_j^{\min}(2 - C_j^t - C_j^{t-1}) \\ P_j^t - P_j^{t+1} \leq (P_j^{down} + \Delta P^{down})(1 - C_j^t + C_j^{t+1}) + P_j^{\min}(2 - C_j^t - C_j^{t-1}) \end{cases} \tag{4}$$

where $P_j^{t+1}, P_j^t$ represent the outputs of coal power unit $j$ in the adjacent moment, respectively; $P_j^{up}, P_j^{down}$ represent the maximum upward and downward ramp rate, respectively.

As a result, Formula (4) can ensure that the output of all the coal power units can be adjusted to any value between the upper and lower boundaries in the next moment. In this way, the coal power units can balance the real-time fluctuations of renewable energy under the described conditions.

### 4.1. Case Data and Scenario Settings

The uncertainty characteristics of WP, PV, and power loads present major challenges to the dispatch operations in the power systems. It has been found that the traditional dispatch operations are no longer suitable for the new dispatch demands of large-scale renewable energy integration. Therefore, highly efficient and schedulable strategies should be combined with the integration of renewable energy and the appropriate traditional resources will need to be determined.

At the present time, PSP plants, electric vehicles, large-scale batteries, and other ES technologies are the most common measures used to restrain the volatility of renewable energy, and have demonstrated certain abilities. Therefore, thermal power, WP, and PSP were combined in this study's unit commitment model. It was found that the optimization simulation results had indicated that the PSP plants were critical to increasing the stability, reliability, and economic efficiency of power systems [38,39,48]. Additionally, the PSP plants were verified to enjoy the advantages of WP integration, which helped to improve the efficiency of the WP [34,49,50]. It had also been as a good complement to the WP for managing the positive and negative energy imbalances over time, and as an effective solution for smoothing the WP fluctuations and reducing the operating costs for the wind-thermal power systems [35,40,51,52]. It was considered that a joint operational approach of PV + WP + PSP, which demonstrated enormous potential for reducing the impacts of renewable energy during the operation of power systems, could be used to facilitate the integration process, as well as increase their share to cover future energy demands [53]. All of the aforementioned references demonstrated that the joint operations of WP and PSP could greatly alleviate WP fluctuations in power grids [54,55]. Therefore, in order to facilitate the integration of WP and PV, this study allocated WP-PSP and PV-ES joint operations for the purpose of inhibiting the uncertainties of renewable energy integration.

In the present research investigation, in accordance with a typical day's power load in a northern area of China (Figure 5), a power system's capacity from a 2018 case study was assumed, and the integrated capacity is shown in Table 4.

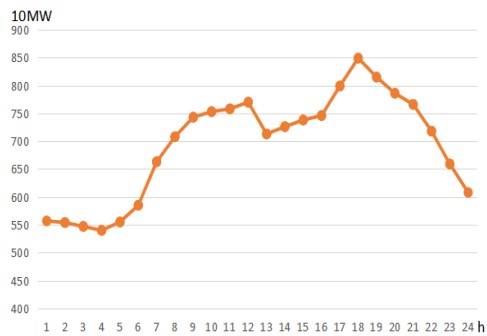

**Figure 5.** A typical day's power load curve.

**Table 4.** Integrated capacities of the examined case.

|  | Total | Coal Power | WP | PV | PSP | ES |
|---|---|---|---|---|---|---|
| **The Examined Case** | 15,365 MW 100% | 7665 MW 49.89% | 4000 MW 26.03% | 3000 MW 19.52% | 400 MW 2.60% | 300 MW 1.95% |
| **Statistical Data from 2018 in China** | - | 53% | 10% | 9% | 1.6% | - |

According to China's power industry statistical data from 2018, the shares of different power resources in this case had conformed to the green development plan of deploying more renewable

energy and containing the coal power installations of China's power industry during the coming years. The concentration of large-scale renewable energy in three northern areas of China were considered, as well as the influences of geographic conditions and water resource constraints. It was found that the share of PSP would not increase by a large percentage. Therefore, its share was set as 2.6% in the examined case. It was presumed that during the next few years, the most promising electric vehicles with large capacity batteries may be widely used in power systems in order to promote the integration of large-scale renewable energy. Therefore, the share of ES was set as 1.95%. Additionally, the shares of WP and PV were set as 26.03% and 19.52%, respectively, in which the output curves were set according to their minimum power unit output curves. As shown in Table 5, three scenarios were set in this study according to the increased capacity shares of the WP and PV.

**Table 5.** Scenarios settings before and after retrofitting.

| Shares of Wind and PV (Renewables) | 21%, 15% (36%) | | 26%, 20% (46%) | | 29%, 23% (52%) | |
|---|---|---|---|---|---|---|
| Ramp Rates | No. | 2 times | No. | 2 times | No. | 2 Times |
| Minimum Output | No. | to 20% rated power | No. | to 20% rated power | No. | to 20% rated power |
| Scenarios | S01 | S02 | S11 | S12 | S21 | S22 |
| | Low share | | BAU | | High share | |

In addition, each scenario was divided into two sub-scenarios before and after retrofitting. It was found that with the changed shares of the WP and PV, both the installed capacities of coal power units, and the ratio of the total installed capacity vs. the maximum load, had remained unchanged.

In the "business as usual" (BAU) scenarios, the examinations of the impacts on the integration of renewable energy with different peak shaving depths following the retrofitting of the coal power units were conducted. Generally speaking, the designed minimum output for IEEE 10 units/39 nodes is approximately 30% of the rated power, and the coal power units usually do not run below a 40% rated power level due to the power system security and economic factors, and so on. Therefore, in order to integrate increased amount renewable energy, five sub-scenarios were set to study the increment integration of renewable energy from 38% of the rated power to 20% of the rated power, as shown in Table 6.

**Table 6.** Scenarios settings after retrofitting with different peak shaving depths.

| Peak Shaving Depth | 38% Rated Power | 35% Rated Power | 32% Rated Power | 30% Rated Power | 20% Rated Power |
|---|---|---|---|---|---|
| Scenarios | S1 | S2 | S3 | S4 | S5 |

### 4.2. Case Study

According to the objective functions and flexible constraint conditions mentioned above, this study operated and analyzed the output curves of the WP and PV based on a CPLEX12.5 solver, which had the ability to contribute to the evaluation of the integration potentials.

### 4.2.1. WP and PV in the BAU Scenarios before and After Retrofitting

As shown in Figure 6, the integrated WP of scenario S12 was higher than that of scenario S11. The joint operation [41] and coal power units flexible operation efforts had successfully restrained the sharp fluctuations of the WP, resulting in 7% reduction of the curtailment rate from scenario S11 to scenario S12. These results had confirmed that the coal power units could promote WP integration through flexible operation strategies.

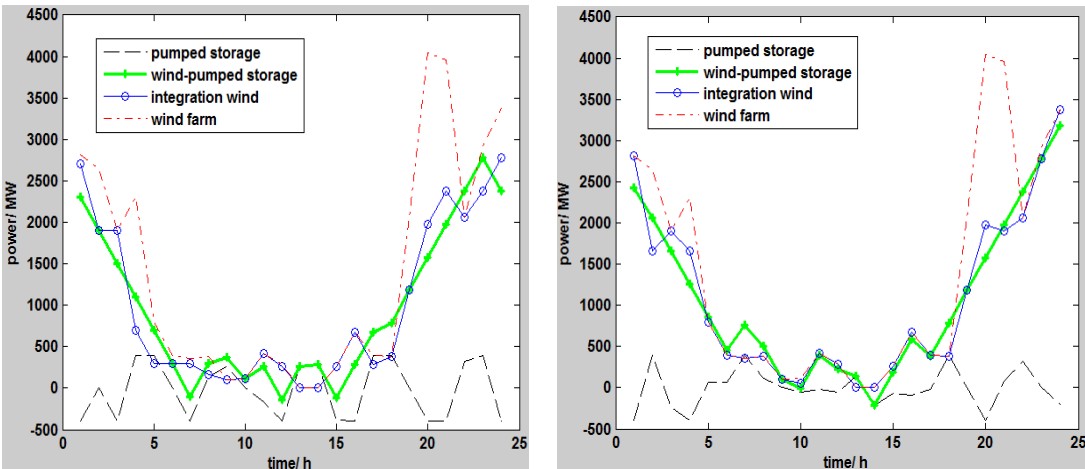

**Figure 6.** Output curves of the wind power (WP) and PSP plant ((**left**) S11, (**right**) S12.)

This study's comparison of scenarios S12 and S11 are displayed in Figure 7. As can be seen in the figure, the power fluctuations had decreased (green line with filled dots) with the ES joint operation efforts and the flexible operations of the coal power units. Meanwhile, the integrated PV in scenario S12 had increased. Furthermore, the PV curtailment rate had decreased 9%, indicating that the flexible operations and joint operation had successfully promoted the integration of the PV [56–58].

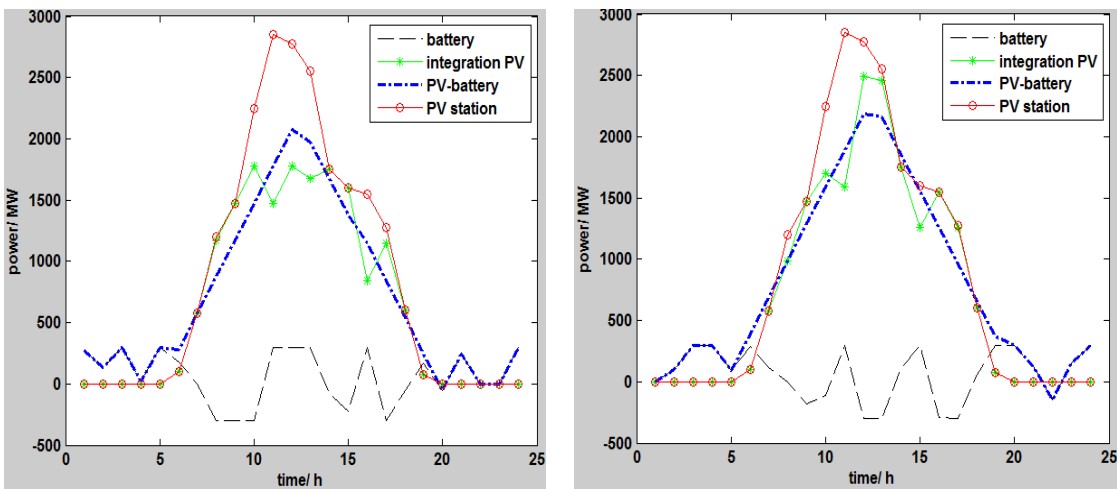

**Figure 7.** Output curves of the photovoltaic power (PV) and ES (Battery) Technologies ((**Left**) S11; (**Right**) S12.)

### 4.2.2. Case Study of Low Share and High Share Scenarios after Retrofitting

It was determined from the flexible operations of the coal power units in the aforementioned three scenarios that the capacity of the renewable energy was the only increased factor. Meanwhile, the capacity of coal power units and the ratio of the total installed capacity vs. the maximum load had remained constant.

As detailed in Table 7, according to the simulation results obtained in scenarios S02 to S12, the share of coal power had decreased 9.5%, while the WP and PV integration had noticeably increased with the flexible operations of the coal power units. The consumption rates of the WP and PV were determined to be 86% and 79%, respectively. Additionally, the integration increment ratios of the WP and PV were approximately 15% and 10%, respectively. As revealed in the results obtained from scenarios S12 to S22, the total capacity share of the WP and PV had increased to 52%, and the capacity of coal power was the same as in scenario S22. At the same time, the consumption rates of the WP

and PV were 77% and 72%, respectively. However, the utilized WP and PV had experienced a total reduction of 16%. Therefore, it was confirmed that with the increasing capacity of the WP and PV, the existing coal power units would not have the ability to supply enough flexible resources to promote their integration. Therefore, additional flexible resources would be necessary to meet the flexibility requirements [10,59–61].

**Table 7.** Consumption rates of the WP and PV.

| Consumption Rates | S02 | S12 | S22 |
|:---:|:---:|:---:|:---:|
| WP | 71% | 86% | 77% |
| PV | 69% | 79% | 72% |

### 4.2.3. Case Study after Retrofitting with Different Peak Shaving Depths

In accordance with the results of the aforementioned BAU scenarios, an estimate of the integration potential with different peak shaving depths of coal power units was also completed in this study. The results are shown in Table 8. It was determined that the minimum output had dropped from 38% to 20% of the rated power. Subsequently, the peak shaving depths had risen from 62% to 80% of the rated power. Then, as a convex function curve, the increment of the integrated WP had showed the largest increasing trend from 35% of the rated power to 32%. Meanwhile that of the PV had displayed a similar trend in the same depth. However, both had displayed a decrescent trend from 32% of the rated power to 20% of the rated power. It was observed that with the increases in the peak shaving depths, the increment potentials of the WP and PV had first increased to 12,125 MW and 11,250 MW, respectively, and then began to decrease to 215 MW and 11 MW, respectively, during which time a transmittance peak had appeared in the interval from a 32% rated power level to a 20% rated power level.

**Table 8.** Integrated WP and PV with different peak shaving depths (unit: MW).

| Scenarios | S1 (38%) | S2 (35%) | S3 (32%) | S4 (30%) | S5 (20%) | Increments | | | |
|:---:|:---:|:---:|:---:|:---:|:---:|:---:|:---:|:---:|:---:|
| | | | | | | S2–S1 | S3–S2 | S4–S3 | S5–S4 |
| **Integrated WP** | 2651 | 10,605 | 22,730 | 25,636 | 25,851 | 7954 | 12,125 | 2906 | 215 |
| **Integrated PV** | 1011 | 4807 | 16,057 | 16,084 | 16,095 | 3796 | 11,250 | 27 | 11 |

In the present study's investigations, it was found that as the coal power units ran at approximately 30% of the rated power with the largest increment flexibility potential, the integrated WP and PV were the highest. This conclusion was also validated by an example in the optimized peak shaving depth, which was based on the residual load for the purpose of estimating the largest flexibility potential in Reference [62]. It was determined that although the coal power units running at approximately 20% of the rated power had the largest peak shaving depth, since the output power was lower than 30% of the rated power, it would in fact induce some faults in the unit's stable-combustion of the units and impact the normal processes of the denitration systems. Therefore, due to the increasing coal consumption and own consumption rates, the low-load operation with approximately 20% of the rated power were determined to be not economical. As a consequence, it was deemed not a feasible solution to operate the existing coal power units at rates lower than 30% of the rated power in China at the present time.

## 5. Conclusions

This study first reviewed the reasons why the flexible operations of the existing coal power units would effectively promote the integration of renewable energy in China. Also, this study elaborated on the potential impacts of the flexible operation practices on the performances of the existing coal power units. Then, using the obtained simulation results, the integrated potential of the coal power units with appropriate flexible operation was estimated.

This study's simulation results showed that the PSP and ES technologies could potentially effectively restrain the uncertainties and severe volatility of the renewable energy. However, the integrated potential was severely impacted by their small capacity of the aforementioned technologies. Therefore, it was recommended that in order to improve the integration of renewable energy in China, the power system was required to more flexibly operate the existing coal power units. For example, it was observed in the simulations that when the total capacity of the WP and PV had progressively increased to a value exceeding 52%, the existing coal power units could not adequately respond to the volatility of renewable energy in a timely manner and under the conditions of power balance constraints. The increment integrated potential at 20% of the rated power for the coal power units was lower than that of the 30% rated power level. Luo et al. had previously found that in the forecast for 2020, the wind curtailment rate would be 15% in northern Hebei, regardless of the economy of the peak shaving. Meanwhile, the minimum technical output of the condensing units would equal to 30% of the rated power, and the 20% minimum technical output had corresponded to a 5% curtailment rate of the wind power [11]. These projections had indicated it would successfully reach the international advanced levels achieved in the majority of the European countries [63]. Therefore, it was recommended that by relying on the flexible operation of the existing coal power units in China, a 5% curtailment of renewable energy could certainly be achieved in the next few years [11].

In addition, the safe and stable operation of the power system is a priority for China's power sector. Therefore, it will be necessary for dispatch operators to curtail the intermittent renewable energy, once the power fluctuations exceed the system's limit. In other words, the power fluctuation constraints of the renewable energy sources will potentially become one of the main bottlenecks for their integration. According to the standards regarding the technical rules for connecting WP and PV technologies into the national power grid [64,65] and State Grid Corporation of China (SGCC) [66,67], the power fluctuations for connecting the WP and PV into the power grid should be limited to between 10% and 33% among the different capacities and time scales. However, according to the local power grid situations, specific values can be given by the dispatch operators which should be allowed to be exceeded [66,67]. Then, once the power fluctuation constraints of the integrated WP and PV become flexible for the different local power grids [68,69], the curtailment of the WP and PV can theoretically be reduced. These steps would provide feasible solutions for the dispatch operators in the practical power system situations and achieve the safe operations of certain power systems.

**Author Contributions:** C.N., H.P. and J.Y. (Jiahai Yuan) conceived this paper; C.N., analyzed the data and wrote the paper; Y.Z. contributed to the revision of the paper; L.D. and J.Y. (Jungang Yu) collected the analysis data.

**Funding:** This paper was funded by: (1) Ningxia key research and development program under Grant No. 2019BDE03007; (2) Ningxia key research and development program (Special Talents) under Grant No. 2018BEB04029; (3) National Natural Science Foundation of China under Grant No. 61763040; (4) Natural Science Foundation of Ningxia University under Grant No. ZR1706.

**Acknowledgments:** The authors thank the anonymous reviewers for their helpful suggestions and comments which improved this work.The responsibility for the contents lies with the authors.

**Conflicts of Interest:** The authors declare no conflict of interest. The founding sponsors had no role in the design of the study; in the collection, analyses, or interpretation of data; in the writing of the manuscript, and in the decision to publish the results.

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
