# Peer review of "The Flexible Operation of Coal Power and Its Renewable Integration Potential in China"

_sustainability, doi:10.3390/su11164424_

Round 1
Reviewer 1 Report
General comment: This paper describes the reasons of coal power unit flexible operation promoting the renewable integration in China.
Introduction: The Introduction should be improve focusing on the aim of the paper, main methods, main results and few recommendations based on empirical results. The authors should explain their research novelty compared to previous studies from literature.
Methodology: Indicate limits and advantages of methods. Indicate alternative methods. Provide practical comments to introduce the methods.
Results: The interpretations are too superficial. More comments of the results are required and comparisons with similar studies from literature. More details on data are required. The contribution is weak. There are too many tables without any relevance.
Discussion: Interpretations of the results are not provided. The literature review is quite superficial.
Bibliography/References: Add more references.
Decision: Accept with corrections.

Author Response
Dear Reviewer,
Great thanks for offering us the opportunity to revise the manuscript. We have referred to the reviewer comments thoroughly and revised the manuscript accordingly. Please see the attachment.
Best regards!
The authors

Reviewer 2 Report
This paper firstly focuses on the reasons of coal power units 16 flexible operation promoting the renewables integration in China, and reviews its impact on the 17 unit’s performances, then a simple flexibility operation model is built to estimate the integration 18 potential with the existing coal power units under several different scenarios. Simulation results 19 show that the existing retrofitted coal power units can provide flexibility to promote the 20 renewables integration in a certain extent, and the integration potential increment of 20% rated 21 power of coal power units is lower than that of 30% rated power.
Even though the paper is well written, it is not clear why, once the power fluctuation constraint of the integrated WP and PV is flexible for different power system, it can reduce the curtailment of WP and PV which would make it one of the feasible solutions in the practical power system for the dispatch operator under the safe condition of a certain power system. It should be explained more in detail.
Furthermore, a much clearer explanation of how the manuscript contributes to the gap of scientific knowledge should be added.
Author Response
Dear Reviewer,
Great thanks for offering us the opportunity to revise the manuscript. Your affirmation to our work gives us a lot encouragement to do a better job. We have revised the manuscript according to the reviewer comments. Please see the attachment.
Best regards!
The authors

Round 2
Reviewer 1 Report
Accept in present form
Author Response
Dear Reviewer,
Great thanks for offering us the opportunity to revise the manuscript. Your affirmation to our work gives us a lot encouragement to do a better job. We have revised English language and style and marked it in the manuscript. Attached is the marked manuscript.
Best regards!
The authors

Reviewer 2 Report
Even though the paper is well written, paper originality/novelty, significance of content, quality of presentation, scientific soundness, interest to the readers, and overall merit should be better explained
Author Response
Dear Reviewer,
Great thanks for offering us the opportunity to revise the manuscript. Your affirmation to our work gives us a lot encouragement to do a better job. We have revised the manuscript according to the reviewer comments, and revise the manuscript by a native English speaking colleague. Attached is the marked manuscript.
Best regards!
The authors
